# Cavitation Prevention Potential of Hydromechanical Pressure Compensation Independent Metering System with External Active Load

**Kailei Liu, Shaopeng Kang \*** , **Hongbin Qiang and Chengtao Yu**

School of Mechanical Engineering, Jiangsu University of Technology, Changzhou 213001, China; liukailei@163.com (K.L.); qianghb@jsut.edu.cn (H.Q.); yuct@jsut.edu.cn (C.Y.)
\* Correspondence: kangshaopengjsut@163.com

**Abstract:** This article studies the cavitation performance and preventing method of the hydromechanical pressure compensation independent metering system (HPCIMS). Compared with the conventional load sensing system (CLSS), the meter-in and meter-out orifices of HPCIMS can be regulated independently. A quasi-static behavior analysis of cavitation performance was applied to the HPCIMS and CLSS. The meter-in pressure equation of HPCIMS showed that keeping the ratio of the meter-in and meter-out orifices greater than the minimum value can avoid the cavitation phenomenon. Systems parameters were then kept as constant, and the key parameters related to cavitation performance of the two systems were compared by varying external force. Comparison results show that the cavitation phenomenon in the meter-in chamber of CLSS with the external active load is inevitable, but in HPCIMS, it can prevent the cavitation phenomenon by changing the ratio of the meter-in and meter-out orifices, so the HPCIMS has the cavitation prevention potential.

**Keywords:** independent metering system; cavitation prevention; external active load; hydraulic-mechanical pressure compensation





## 1. Introduction

Due to the unique and valuable characteristics, hydraulic systems are widely used for a variety of applications ranging from construction to industrial, military, aerospace, and earth moving applications [1,2]. The mobile hydraulic system, which is one of the typical hydraulic systems, has many demanding requirements, like changing environment, limited space, and higher power/weight, and so on [3]. Generally, the proportional directional spool valves are widely used in the mobile hydraulic system to control the flow direction and flow rate of the actuator [4]. A sliding spool is adopted in the proportional directional spool valve, so the meter-in orifice and meter-out orifice are mechanically linked. For example, the conventional load sensing system is a typical mobile hydraulic system, which is widely used in construction machinery and agricultural machinery. The proportional directional spool valve is used in the conventional load sensing system, which makes the system easy to control, but it also brings in some significant limitations, such as energy losses and nonlinearities, which make the control system more challenging [5]. In the conventional mobile load sensing system, about 30% of the energy losses are due to the friction and hysteresis of the proportional directional spool valves [6].

In order to overcome the shortcomings of the traditional hydraulic system, A. Jansson and J. O. Palmberg proposed a hydraulic system using four 2/2-valves to control an actuator independently; it breaks the mechanical linkage and decouples the restriction [7]. In this system, the motion control of the actuator can be realized by actuating two of the four 2/2-valves. Compared with the conventional load sensing system, the independent metering configuration has brought many functions due to the decoupling of meter-in and meter-out orifices, including regeneration functionality, float functionality, energy-saving

functionality, and cavitation prevention functionality [8]. The development of independent metering technology is technically and economically attractive, and it can replace the traditional hydraulic technology [9]. The independent metering technology can be realized by many available valve configurations, for example, five 2/2-valves, two 3/3-valves, a combination of two 3/3-valves and one 2/2-valve [10]. Different valve configurations require different control logics and corresponding control methods for desired actuator motion control. Many previous researchers have investigated the independent metering technology. B. Yao adopted five electro-hydraulic proportional cartridge valves complete trajectory tracking and energy-saving control by robust adaptive control algorithm [11–13]. Hu H and Zhang Q using a programmable E/H valve with a hybrid control algorithm, realize the multifunction [14,15]. Moreover, good energy-saving characteristics of excavators with the independent metering system must be obtained by using different control methods [16–18].

Compared with the conventional hydraulic system, the independent metering system is a multiple-input and multiple-output system, so it requires a more complex control method for motion and displacement control. In the mobile hydraulic system, the motion of the actuator needs to be consistent with the input signal of a joystick, so the flow rate passing through the main control valve must be controlling. There are two control methods for flow rate control: the electro-hydraulic pressure compensation method and the hydromechanical pressure compensation method. The electro-hydraulic pressure compensation method needs more sensors and a complex control algorithm for realizing the motion control. As opposed to the electro-hydraulic pressure compensation method, the hydromechanical pressure compensation method reacts directly and rapidly to disturbance variables in the system due to using the hydraulic pressure compensator [19]. Compared with the traditional load sensing system, the load sensing system with hydromechanical pressure compensation and independent metering offers more significant energy savings [20].

Compared with the conventional hydraulic system, the independent metering system has a very important function is that can prevent the cavitation phenomenon by regulating the meter-in and meter-out orifices. As well known, the cavitation phenomenon takes place due to local evaporation of the fluid caused by a lowering of the pressure. If the pressure falls below the vapor pressure of the fluid, the steam bubble is created, and the bubbles reach regions of higher pressure, it will be generated pressure waves and caused erosion [21]. The cavitation may occur in the pumps, valves, actuators, and pipes of the hydraulic system, and it will result in severe damage to the hydraulic components and the whole hydraulic system [22–25].

In the hydraulic system, there are two kinds of external load: external passive load and external active load. With the external passive load, the motion direction of the actuator is opposite to the direction of the external load; it will hinder the movement of the actuator. In the meter-in chamber and meter-out chamber of the actuator with external passive load, it will not be easy to produce cavitation. With the external active load, the motion direction of the actuator is the same as the direction of the external load, and it will drive the movement of the actuator. If the external active load is great than a certain value, the pressure of the meter-in chamber of the actuator will fall below the vapor pressure, as a result, will be produced the cavitation phenomenon. In the conventional load sensing system with greater external active load, the cavitation phenomenon is inevitable. However, in the independent metering system, the cavitation phenomenon can be avoided by regulating the meter-in and meter-out orifices [10].

Many scholars have studied the independent metering system for a long time, but the research related to the cavitation performance analysis of the independent metering system is still very rare. This study focuses on the quasi-static behavior of the hydromechanical pressure compensation independent metering system with external active load and ignores the dynamic effects. Cavitation performance and preventing method of the hydromechanical pressure compensation independent metering system are investigated in this paper.

The remainder of this paper is organized as follows: Section 2 introduces the main layouts and the working principles of the hydromechanical pressure compensation independent metering system and conventional load sensing system, respectively. Section 3 presents the cavitation performance analysis of the hydromechanical pressure independent metering system and conventional load sensing system. The quasi-static behavior equations of the two systems with external active load are deduced in detail in Section 3, as well as the preventing cavitation method of the hydromechanical pressure independent metering system is proposed. Section 4, system parameters are kept constant, and the cavitation performances of the hydromechanical pressure independent metering system and conventional load sensing system are presented for the sake of comparison. Section 5 concludes the paper by discussing the cavitation prevention applicability and further work of the hydromechanical pressure independent metering system.

## 2. System Layouts and Working Principle

### 2.1. Working Principle

The principle of a spool valve controls a cylinder is shown in Figure 1. As is shown in Figure 1, the sliding spool $x$ of the spool valve plays an important role in the system that can control the flow direction and the flow rate. When the sliding spool $s$ is moving to the left location, the high-pressure oil flows from the port P of the spool valve to the head chamber of the cylinder, the flow rate of metering in the orifice is $Q_a$, and then the high-pressure oil pushes the cylinder piston rod to extend. As a result, the low-pressure oil flows the rod chamber of the cylinder to the port T of the spool valve, the flow rate of metering out orifice is $Q_b$. When the sliding spool $s$ is moving to the right location, the cylinder piston rod can be realized retraction motion.

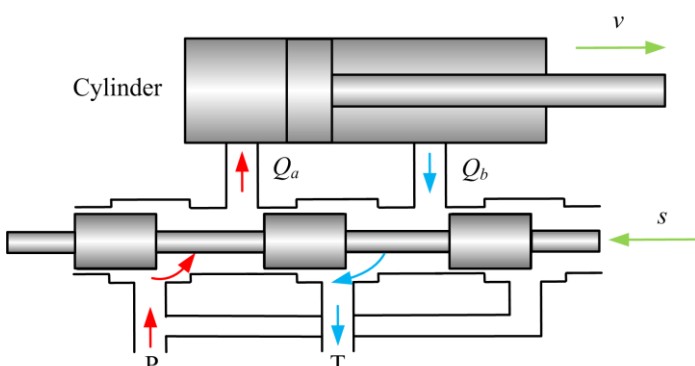

**Figure 1.** Principle of a spool valve controls a cylinder.

It can be seen that the flow rate of meter-in orifice $Q_a$ and the flow rate of meter-out orifice $Q_b$ are controlled simultaneously by the sliding spool $s$. However, the flow rate of meter-in orifice $Q_a$ can determine the velocity $v$ of the cylinder piston rod, so the decrease of the meter-out orifice by the sliding spool motion is redundant because it caused repeated throttling losses.

To overcome the shortcomings of the conventional spool valve control system, the independent metering control system is has appeared. The principle of the independent metering control system is shown in Figure 2. It can be seen that the spool valve of the conventional spool valve control system is replaced by four spool valves #1, #2, #3 and #4. When the sliding spool $s_1$ is moving to the left location, the high-pressure oil flows from the shared port P of the spool valve #1 and #3 to the head chamber of the cylinder. Then the high-pressure oil pushes the cylinder piston rod to extend, and simultaneously, the sliding spool $s_4$ regulates the meter-out orifice. As a result, the low-pressure oil flows out of the rod chamber of the cylinder to the share port T of the spool valve #2 and #4. The flow rate of the meter-in orifice of spool valve #1 is $Q_a$, and the flow rate of the meter-out orifice

of spool valve #4 is $Q_b$. When the sliding spool $s_3$ and $s_2$ are moving to the left location, the cylinder piston rod can be realized retraction motion.

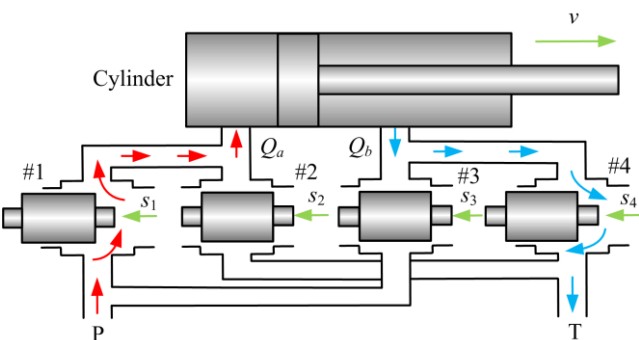

**Figure 2.** Principle of an independent metering control system.

Hence, compared with the conventional spool valve control system, the independent metering system has two control degrees whether the cylinder piston rod is extending or retracting. When the flow rate of metering in the orifice of spool valve #1 is controlled by the sliding spool $s_1$, the sliding spool $s_4$ can move to the maximum position to reduce the repeated throttling losses as soon as possible. Finally, it can achieve energy saving.

There are many hydraulic system layouts that can realize the independent metering system. Figure 3 shows the two typical kinds of hydraulic system layouts. Figure 3a shows the four 2/2-valves control system layouts. Figure 3b shows the two 3/3-valves controlled system layouts. However, the two typical hydraulic system layouts have different kinds of valves and various connection formations, but they both can reduce the repeated throttling losses for energy saving.

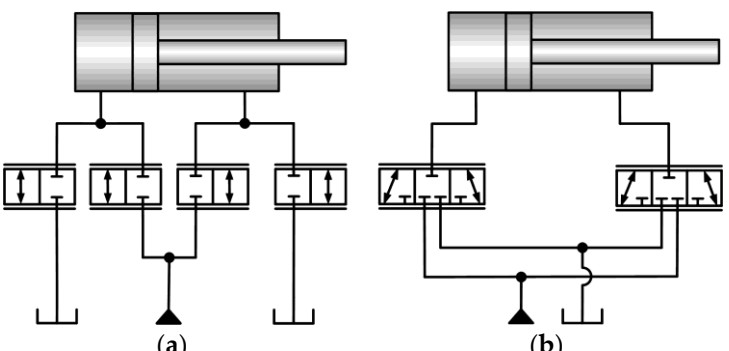

**Figure 3.** Two typical kinds of hydraulic system layouts. (**a**) Four 2/2-valves (**b**) two 3/3-valves.

*2.2. Electro-Hydraulic Pressure Compensation Method*

In the independent metering system, the flow rate $Q$ of the spool valve can also be expressed:

$$Q = C_d W x \sqrt{\frac{2\Delta P}{\rho}} \tag{1}$$

where $C_d$ is the flow coefficient of the spool valve, $W$ is the area gradient of the spool valve, $x$ is the sliding spool displacement of the spool valve, $\Delta P$ is the differential pressure of the spool valve, and $\rho$ is the density of the hydraulic fluid.

From Equation (1), the density of the hydraulic fluid $\rho$ can be seen as a constant value. The parameter $W$ is the area gradient of the spool valve, which is determined by the dimensions of the spool valve, so it can seem like a constant value when the spool valve is the same. The flow coefficient $C_d$ equals 0.611 in the turbulent region [26], so it can be seen as a constant value because the turbulent flow existed in the valve control

system. Hence, the differential pressure $\Delta P$ and the sliding spool displacement $x$ can be changed for the desired flow rate $Q$. As a result, that obtaining the desired flow rate $Q$ of the spool valve, there are two methods that can be accomplished. One method is called the electro-hydraulic pressure compensation method, which is changing the sliding spool displacement $x$ by the electronic signal according to the differential pressure $\Delta P$. Moreover, the other method is called the hydromechanical pressure compensation method, which is keeping the differential pressure $\Delta P$ as a constant value, change the sliding spool displacement $x$ simultaneously.

The principle of the electro-hydraulic pressure compensation method is shown in Figure 4. The inlet valve and the outlet valve have separately controlled the flow rate of metering in orifice $Q_a$ and the flow rate of metering out orifice $Q_b$. Three pressure sensors should be existed for detecting the pressure $P_s$, $P_a$, and $P_b$; the pressure $P_0$ is approximately regarded as 0 because it is the tank pressure. In order to obtain the desired flow rate $Q$, it is necessary to control sliding spool displacement $x$ by closed-loop feedback according to Equation (1), so two-spool displacement sensors should have existed.

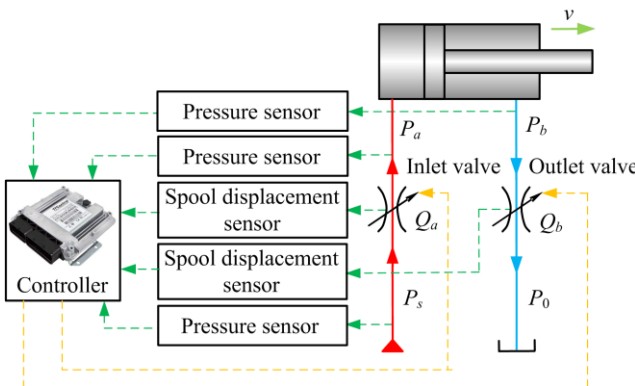

**Figure 4.** Electro-hydraulic pressure compensation method.

The advantage of the electro-hydraulic pressure compensation method is that it reduces the repeated throttling losses because the inlet valve and outlet valve can be separately controlled, but the disadvantages, like more sensors needed, diagnostic points added, and closed-loop feedback control demanding, also has been brought out.

*2.3. Hydromechanical Pressure Compensation Method*

The principle of the hydromechanical pressure compensation method is shown in Figure 5. The pressure compensator, which is a hydraulic component, was adopted for keeping the differential pressure $\Delta P$ of the inlet valve as a constant value. Hence, from Equation (1), the density of the hydraulic fluid $\rho$, the area gradient $W$, and the flow coefficient $C_d$ can be seen as constant values, when the differential pressure $\Delta P$ is constant due to the pressure compensator, the desired flow rate $Q$ is proportional to the sliding spool displacement $x$. As a result, the spool displacement sensors are not required, and only two pressure sensors are needed for detecting the pressure $P_a$ and $P_b$. Thus, the controller no need to detect the differential pressure $\Delta P$ and the sliding spool displacement $x$; it only needs to send control signals to the inlet valve and outlet valve separately.

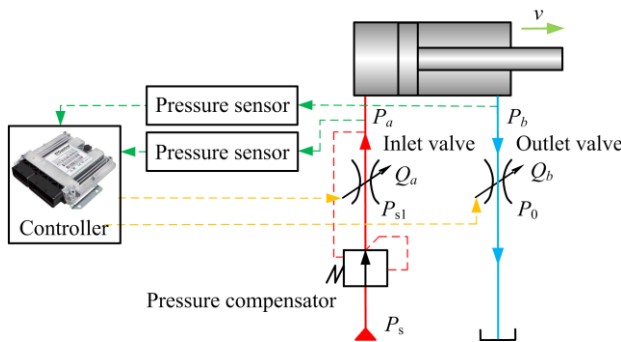

**Figure 5.** Hydromechanical pressure compensation method.

Compared with the electro-hydraulic pressure compensation method, the hydromechanical pressure compensation method reduced one pressure sensor and two spool displacement sensors, so the failure rate of the control system will be lower. Moreover, the pressure compensator can keep the differential pressure $\Delta P$ of the inlet valve as a constant by the hydromechanical principle; it is more stable for the control system. However, the disadvantage of the hydromechanical pressure compensation method is that the pressure compensator brings a certain pressure loss.

### 2.4. HPCIMS Configuration

According to the principle of the hydromechanical pressure compensation method, the hydromechanical pressure compensation independent metering system (HPCIMS) can be designed. There are many configurations that can realize the HPCIMS, one of the configurations as is shown in Figure 6. The hydromechanical pressure compensation independent metering control block, which is integrating multiple valves, including five 2/2-valves, two switch 3/2-valves, a pressure compensator and a shuttle valve. The main role of the five 2/2-valves is to complete the extension and retraction motion of the cylinder. The pressure compensator can keep the differential pressure of the 2/2-valve #4 or 2/2-valve #5 as a constant value. The combination of the switch 3/2-valve #1, switch 3/2-valve #2 and shuttle valve can truly feedback the pressure of the two chambers of the cylinder to the port $LS_1$.

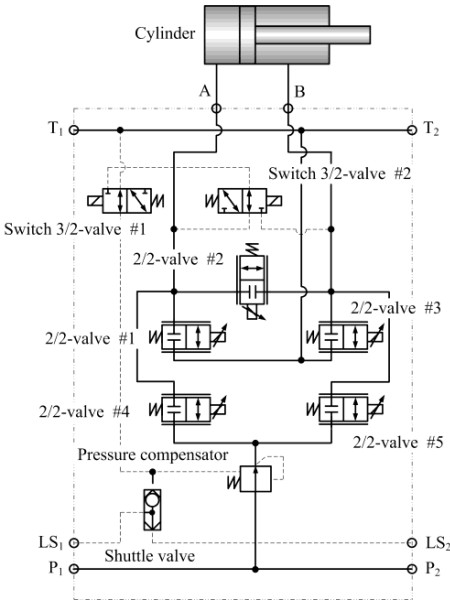

**Figure 6.** Hydromechanical pressure compensation independent metering control block controls a cylinder.

*2.5. CLSS Configuration*

The conventional load sensing system (CLSS) is commonly used in construction machinery and agricultural machinery. Figure 7 shows the conventional load sensing control block that controls a cylinder. Compared with HPCIMS, the CLSS also has adopted a proportional directional spool 5/3-valve, a pressure compensator, a shuttle valve, and two proportional pressure reducing valves. The differential pressure of the proportional directional spool 5/3-valve is kept as a constant value by the pressure compensator. The proportional directional spool 5/3-valve has completed the motion direction control of the cylinder piston rod. The proportional pressure reducing valves #1 and #2 can control the sliding spool displacement. The pressure of the two chambers of the cylinder can feedback to the port $LS_1$ by the proportional directional spool 5/3-valve and shuttle valve.

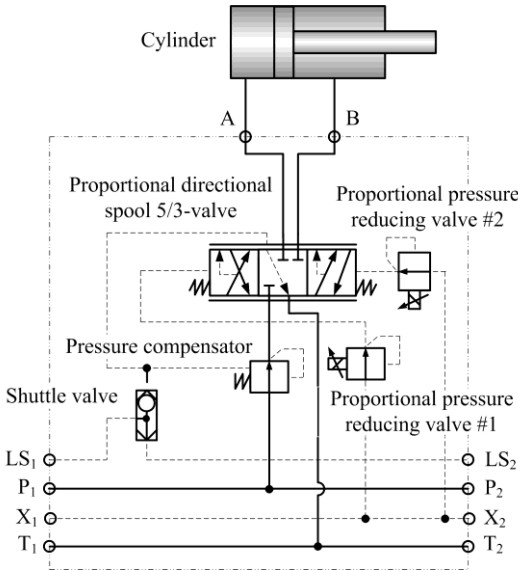

**Figure 7.** Conventional load sensing control block controls a cylinder.

## 3. Cavitation Performance Analysis

### 3.1. Causes of Cavitation

The cylinder is the important hydraulic component in the hydraulic system; it can complete retraction and extension movements. When the piston of the cylinder is moving, the external force acting on the piston rod has two forms: external passive force and external active force. In actual applications of the cylinder, both passive and active load forces may be excessive. The consequence of excessive external passive load is that the piston rod of the hydraulic cylinder cannot move, which will cause the mechanical drive system to suspend. The consequence of excessive external active load is that one of the cylinder chambers will produce cavitation.

In the hydraulic system, the system pressure is generally the gauge pressure. As is shown in Figure 8, cavitation is a phenomenon, which can occur in the hydraulic system, such as pumps, valves, motors and cylinders, when the hydraulic system pressure falls below the vapor pressure $P_c$, which is the gauge pressure. The hydraulic system pressure area is divided into three areas: the working pressure area, cavitation pressure area and vacuum pressure area. It can be seen that the working pressure area is desired, as the hydraulic system pressure is greater than the vapor pressure $P_c$. Moreover, the cavitation pressure area is included the vacuum pressure; both of the cavitation pressure areas and vacuum pressure are not desired. As the hydraulic system pressure is less than the vapor pressure $P_c$, it will produce cavitation. If the hydraulic system pressure is less than 0, it will produce a vacuum, and the minimum value of the system pressure is the negative atmospheric pressure $-P_{at}$.

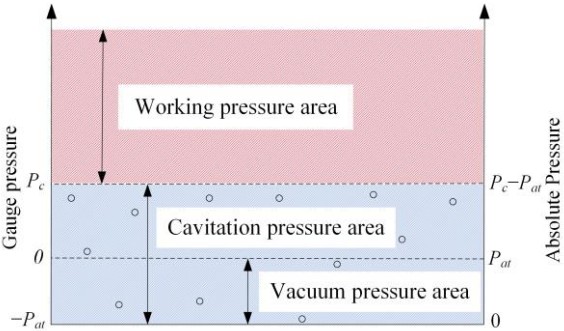

**Figure 8.** The hydraulic system pressure area division.

In the hydraulic system with external active load, both the extension working mode and retraction working mode will produce cavitation. Figure 8 shows two working modes with external active load. Figure 9a shows the extension working mode with external active load, and Figure 9b shows the retraction working mode with external active load. From Figure 9a,b, it can be seen that the direction of the external active load $F_L$ acting on the piston rod is the same as the direction of the cylinder piston speed $v$. The difference between extension working mode and retraction working mode is that the direction of the velocity $v$, the flow rate $Q_a$ and $Q_b$, and the force $F_L$ and $F_P$ are opposite. Hence, take the extension working mode with external active load, for example, for analyzing the cavitation phenomenon.

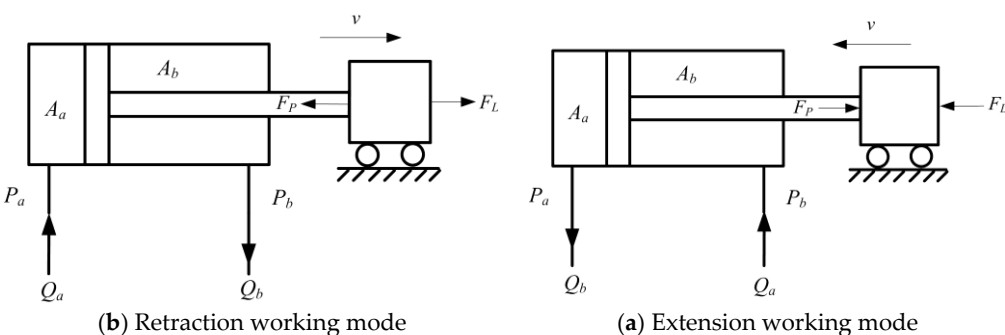

(**b**) Retraction working mode        (**a**) Extension working mode

**Figure 9.** Two working modes with external active load.

As is shown in Figure 9a, the output force $F_P$ of the piston rod can be express:

$$F_P = P_b A_b - P_a A_a \tag{2}$$

where $F_P$ is the output force of the piston rod, $A_a$ is the head chamber area of the cylinder, $A_b$ is the cylinder rod chamber area, $P_a$ is the cylinder head chamber pressure, $P_b$ is the cylinder rod chamber pressure.

As is shown in Figure 9a, when the piston rod is extending with the external active load $F_L$, the output force $F_P$ will be hindering the extension motion of the piston rod. The value of output force $F_P$ depends on the head chamber pressure $P_a$ and rod chamber pressure $P_b$. If the rod chamber pressure $P_b$ is growing to the maximum safe pressure of the hydraulic system, and the head chamber pressure $P_a$ the negative atmospheric pressure $-P_{at}$. As a result, the output force $F_P$ will be growing to the maximum $F_{Pmax}$. Hence, in the hydraulic system, the output force $F_P$ must be less than the maximum out force $F_{Pmax}$.

When the external active load $F_L$ acting on the piston rod is less than the maximum out force $F_{Pmax}$, the force balance equation for the piston rod can be expressed:

$$F_L = F_P \tag{3}$$

In this situation, as Equation (3) is described, the velocity $v$ of the cylinder piston rod is controlled by the hydraulic system, and it will not produce the cavitation phenomenon in the head chamber of the cylinder.

When the external active load $F_L$ is greater than the maximum out force $F_{Pmax}$, the force balance equation for the piston rod can be expressed:

$$F_L - F_P = ma \tag{4}$$

where $F_L$ is the external active load acting on the piston rod, $m$ is the total mass of the piston and load referred to the piston, $a$ is the acceleration of the piston and load referred to the piston.

The velocity $v$ of the piston can be expressed:

$$v = v_0 + at \tag{5}$$

where $v_0$ is the velocity when the external active load $F_L$ equals the output force $F_P$ of the piston rod, $t$ is the acceleration time.

In this situation, as Equations (4) and (5) are described, the velocity $v$ of the cylinder piston rod will be increasing with the acceleration time $t$. If the acceleration time $t$ lasts too long, the velocity $v$ will be increasing from velocity $v_0$ to a great value $v_{max}$. Hence, the velocity $v$ of the cylinder piston rod cannot be depending on the hydraulic system but on the external active load $F_L$ and the acceleration time $t$.

In the two working modes with external active load, if both of the cylinder rod chamber pressure $P_b$ and the cylinder head chamber pressure $P_a$ are greater than 0, which is the gauge pressure, the flow rate $Q_a$ and $Q_b$ can be calculated by the continuity equations:

$$Q_a = A_a v \tag{6}$$

$$Q_b = A_b v \tag{7}$$

where $Q_a$ is the flow rate of the cylinder head chamber, $Q_b$ is the flow rate of the cylinder rod chamber.

In the extension working mode with the eternal, active load as is shown in Figure 9a. From Equation (6), when the velocity $v$ is increasing to a great value $v_{max}$, the inflow rate $Q_a$ of the cylinder head chamber will be increased to a great value $Q_{amax}$. However, the output flow rate $Q_p$ of the pump is fixed. If the $Q_{amax}$ is greater than the output flow rate $Q_p$ of the pump, the cylinder head chamber will produce cavitation. As a result, the flow rate of the cylinder head chamber $Q_a$ cannot be calculated by Equation (6). However, $Q_b$ is the outflow rate of the cylinder rod chamber, so it cannot produce cavitation, as a result, that the flow rate $Q_b$ can be calculated by Equation (7).

The retraction working mode with external active load is the same as the extension working mode with eternal, active load. As is shown in Figure 9b, when the external active load $F_L$ is greater than the maximum out force $F_{Pmax}$, the cylinder rod chamber will be produced the cavitation phenomenon, as a result, that the flow rate $Q_b$ cannot be calculated by Equation (7). Similarly, the flow rate $Q_a$ can be calculated by Equation (6).

*3.2. HPCIMS*

In this section, the fluid compressibility and the dynamic effects are also ignored. HPCIMS, as is shown in Figure 6, can be simplified as is shown in Figure 10.

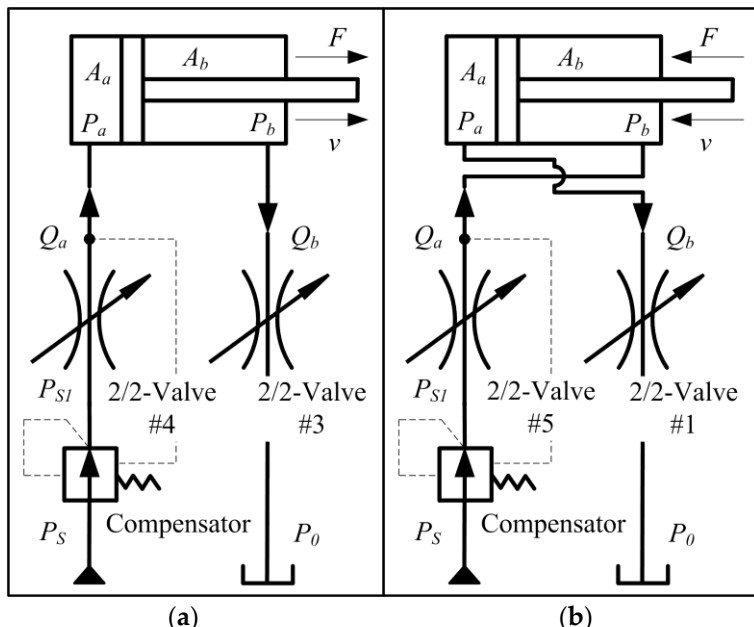

**Figure 10.** Simplified schematic of hydromechanical pressure compensation independent metering system (HPCIMS) with two working modes. (**a**) Extension working mode, (**b**) retraction working mode.

Figure 10a shows the simplified schematic of HPCIMS with the extension working mode, and it can be seen that the 2/2-valve #4 is controlling the inflow rate $Q_a$ of the cylinder head chamber, and the 2/2-valve #3 is controlling the outflow rate $Q_b$ of the cylinder rod chamber; as a result, the cylinder piston rod can be extended with the external active load $F_L$. Figure 10b shows the simplified schematic of HPCIMS with the retraction working mode; it can be seen that the 2/2-valve #5 is controlling the inflow rate $Q_a$ of the cylinder rod chamber, and the 2/2-valve #1 is controlling the outflow rate $Q_b$ of the cylinder head chamber. As a result, the cylinder piston rod can be retracted with the external active load $F_L$. Hence, take the extension working mode, for example, and it is similar to the retraction working mode.

Assume the intrinsic parameters of the 2/2-valve #1, #2, #3, #4, and #5 are the same. As is shown in Figure 9a, flows in 2/2-valve #4 and 2/2-valve #3 can be expressed:

$$Q_a = C_d W x_{\max} x_{in} \sqrt{\frac{2(P_{S1} - P_a)}{\rho}} \tag{8}$$

$$Q_b = C_d W x_{\max} x_{out} \sqrt{\frac{2(P_b - P_0)}{\rho}} \tag{9}$$

where $C_d$ is the flow coefficient of the 2/2-valve, $W$ is the area gradient of the 2/2-valve, $x_{\max}$ is the maximum displacement of the 2/2-valve sliding spool, $x_{in}$ is the opening ratio of 2/2-valve #4, $x_{out}$ is the opening ratio of 2/2-valve #3, $P_{S1}$ is the outlet pressure of the compensator, $P_0$ is the pressure of the tank, and $\rho$ is the density of the hydraulic fluid.

If the external active load $F_L$ is increasing to a big value, the head chamber pressure $P_a$ will be decreased to a negative value. If the head chamber pressure $P_a$ is less than the vapor pressure $P_c$, the head chamber will produce cavitation.

Hence, if the head chamber pressure $P_a$ is greater than the negative atmospheric pressure $-P_{at}$, which is the absolute pressure 0. Take the Equations (2) and (3) into derivation; the results can be obtained:

$$F_L = P_b A_b - P_a A_a \tag{10}$$

If the pressure $P_a$ is less than negative atmospheric pressure $-P_{at,}$ which is the absolute pressure 0, it equals $-P_{at}$, so the external active load $F_L$ can be expressed:

$$F_L = P_b A_b + P_{at} A_a \tag{11}$$

If the pressure $P_a$ is greater than 0, flows in 2/2-valve #4 and 2/2-valve #3 with extension working mode can also be characterized by the following equations:

$$Q_a = A_a v \tag{12}$$

$$Q_b = A_b v \tag{13}$$

Taking the Equations (8), (9), (12) and (13) into derivation, and the results can be obtained:

$$\frac{Q_a}{Q_b} = \frac{x_{in} \sqrt{P_{S1} - P_a}}{x_{out} \sqrt{P_b - P_0}} = \frac{A_a}{A_b} \tag{14}$$

The differential pressure of 2/2-valve #4 is determined by the compensator, so it can be expressed:

$$\Delta P = P_{S1} - P_a \tag{15}$$

By defining $\mu = \frac{x_{in}}{x_{out}}$, $R = \frac{A_a}{A_b}$ and the $P_0$ is the pressure of tank can be assumed as 0. Squaring Equation (14) and rearranging the following expressions, the results are obtained:

$$P_b = \frac{\mu^2}{R^2} \Delta P \tag{16}$$

If the head chamber pressure $P_a$ is greater than negative atmospheric pressure $-P_{at}$, which is the absolute pressure 0, take the Equations (10) and (16) into derivation, the pressure $P_a$ can be obtained:

$$P_a = \frac{\mu^2}{R^3} \Delta P - \frac{F_L}{A_a} \tag{17}$$

If the head chamber pressure $P_a$ is less than negative atmospheric pressure $-P_{at,}$ which is the absolute pressure 0, the pressure $P_a$ equals $-P_{at}$, and $P_b$ can be expressed:

$$P_b = \frac{F_L - P_{at} A_a}{A_b} \tag{18}$$

If the cylinder head chamber pressure $P_a$ is less than the vapor pressure $P_c$, the head chamber of the cylinder will produce cavitation. Hence, the pressure $P_a$ must be greater than the vapor pressure $P_c$. As the opening ratio $x_{in}$ of 2/2-valve #4 and the opening ratio $x_{out}$ of 2/2-valve #3 can be independently regulated, so the ratio $\mu$ is a variable parameter. The ratio $\mu_{min}$ can be expressed:

$$\mu_{min} = \sqrt{\frac{(F_L + P_c A_a) R^3}{A_a \Delta P}} \tag{19}$$

Using the same method, the results of the retraction working mode can be obtained as follows:

$$P_a = \mu^2 R^2 \Delta P \tag{20}$$

If the rod chamber pressure $P_b$ is greater than the negative atmospheric pressure $-P_{at}$, the pressure $P_b$ can be expressed:

$$P_b = \mu^2 R^3 \Delta P - \frac{F_L}{A_b} \tag{21}$$

If the rod chamber pressure $P_b$ is less than negative atmospheric pressure $-P_{at}$, the pressure $P_b$ equals $-P_{at}$, the pressure $P_a$ can be expressed:

$$P_a = \frac{F_L - P_{at}A_b}{A_a} \tag{22}$$

Similarly, if the cylinder rod chamber pressure $P_b$ is less than the vapor pressure $P_c$, the rod chamber of the cylinder will produce cavitation. Hence, the pressure $P_b$ must be greater than the vapor pressure $P_c$. Hence, the ratio $\mu_{min}$ can also be expressed:

$$\mu_{min} = \sqrt{\frac{F_L + P_c A_b}{A_b \Delta P R^3}} \tag{23}$$

In the extension working mode, the pressure $P_a$ is the meter-in pressure, and the pressure $P_b$ is the meter-out pressure. In the retraction working mode, the pressure $P_a$ is the meter-out pressure, and the pressure $P_b$ is the meter-in pressure.

Hence, the meter-in pressure $P_a$ of extension working mode and the meter-in pressure $P_b$ of retraction working mode can be calculated by the Equations (17) and (21). It is clear that the parameters $R$, $A_a$, $A_b$ and $\Delta P$ are constant values. Hence, the meter-in pressure $P_a$ of extension working mode and the meter-in pressure $P_b$ of retraction working mode are related to the variable parameters $\mu$ and $F_L$.

The meter-out pressure $P_b$, which is the cylinder head chamber pressure of extension working mode, can be calculated by Equations (16) and (18). It is clear that the pressure $P_b$ is related to the variable parameter $\mu$ when the head chamber pressure $P_a$ is greater than negative atmospheric pressure $-P_{at}$, and the pressure $P_b$ is related to the variable parameter $F_L$ when the head chamber pressure $P_a$ is less than negative atmospheric pressure $-P_{at}$. The meter-out pressure $P_a$ of retraction working mode can be calculated by the Equations (20) and (22), it is clear the pressure $P_a$ is related to the variable parameter $\mu$ when the rod chamber pressure $P_b$ is greater than negative atmospheric pressure $-P_{at}$, and the pressure $P_q$ is related to the variable parameter $F_L$ when the rod chamber pressure $P_b$ is less than negative atmospheric pressure $-P_{at}$.

Because the meter-in valve opening ratio $x_{in}$ and meter-out opening ratio $x_{out}$ can be regulated independently, the ratio $\mu$ can be seen as an input variable parameter that can change the performance of HPCIMS. Equations (19) and (23) show that the minimum value of the ratio $\mu$ is related to the variable parameter $F_L$.

*3.3. CLSS*

CLSS, as is shown in Figure 7, can be simplified as is shown in Figure 11. Figure 11a shows the simplified schematic of CLSS with the extension working mode. It can be seen that the proportional directional spool 5/3-valve is controlling the inflow rate $Q_a$ of the cylinder head chamber and the outflow rate $Q_b$ of the cylinder rod chamber simultaneously, as a result, that the cylinder piston rod can be extended with the external active load $F_L$. Figure 11b shows the simplified schematic of CLSS with the retraction working mode; it can be seen that the proportional directional spool 5/3-valve is controlling the inflow rate $Q_a$ of the cylinder rod chamber and the outflow rate $Q_b$ of the cylinder head chamber simultaneously, as a result, that the cylinder piston rod can be retracted with the external active load $F_L$. Because the meter-in orifice and the meter-out orifice of the proportional directional spool 5/3-valve are mechanically connected, so the ratio $\mu$ between the meter-in orifice and meter-out orifice is a constant value. In the CLSS, the proportional directional spool 5/3-valve is symmetrical or asymmetrical, which is depending on the designer. The proportional directional symmetrical spool 5/3-valve is commonly used in the CLSS, which means the maximum spool displacements of the meter-in and meter-out are the same. As the mechanical connection of metering edges, so in this section, the proportional directional spool 5/3-valve the valve is symmetrical or asymmetric.

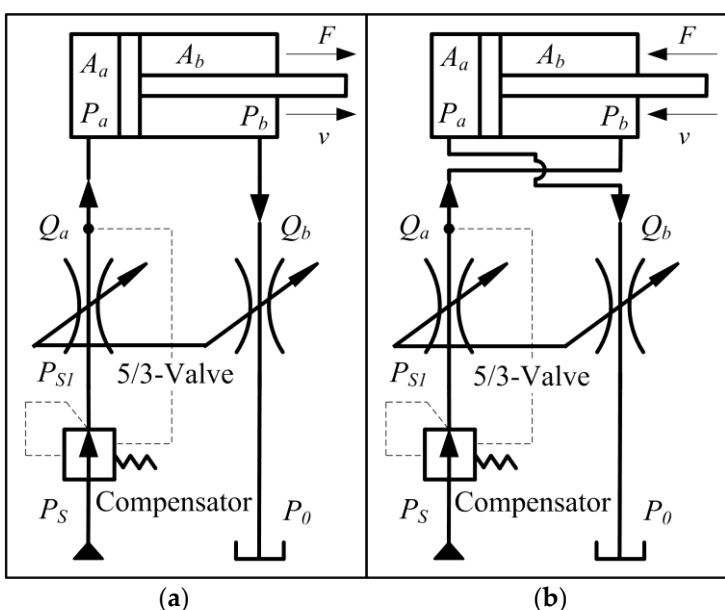

**Figure 11.** Simplified schematic of conventional load sensing system (CLSS) with two working modes. (**a**) Extension working mode, (**b**) retraction working mode.

In the extension working mode, if the head chamber pressure $P_a$ which is the meter-in pressure, is greater than the negative atmospheric pressure $-P_{at}$, the meter-in pressure $P_a$ and the meter-out pressure $P_b$ of the CLSS can be calculated by Equations (16) and (17). It is different from the HPCIMS; the ratio $\mu$ is fixed, which depends on the control valve. The velocity $v$ of the cylinder piston rod can be calculated by Equation (13).

Moreover, if the head chamber pressure $P_a$ is less than the negative atmospheric pressure $-P_{at}$, the pressure $P_a$ equals $-P_{at}$, and $P_b$ can also be calculated by Equation (18).

In the retraction working mode, if the rod chamber pressure $P_b$ which is the meter-in pressure, is greater than the negative atmospheric pressure $-P_{at}$, the equations of the meter-out pressure $P_a$ and the meter-in pressure $P_b$ of the CLSS with retraction working mode can be calculated by Equations (20) and (21). It is different from the HPCIMS; the ratio $\mu$ is fixed, which is depending on the control valve. The velocity $v$ of the cylinder piston rod can be calculated by Equation (12).

If the rod chamber pressure $P_b$ is less than the negative atmospheric pressure $-P_{at}$, the pressure $P_b$ equals $-P_{at}$, the pressure $P_a$ can also be calculated by Equation (22).

The meter-in pressure $P_a$ of extension working mode can be calculated by Equation (17). The meter-in pressure $P_b$ of working retraction mode can be calculated by Equation (21). From the two equations, it is clear that the parameters $\mu$, $R$, $A_a$, $A_b$, and $\Delta P$ are constant. Hence, the meter-in pressure $P_a$ of extension working and the meter-in pressure $P_b$ of retraction working mode are related to the external active load $F_L$.

The meter-out pressure $P_b$ with extension working mode can be calculated by Equations (16) and (18). From the two equations, it is clear that the pressure $P_b$ is a constant when the meter-in pressure $P_a$ is greater than the pressure $-P_{at}$, and the pressure $P_b$ is related to the variable parameter $F_L$ when the meter-in pressure $P_a$ is less than the pressure $-P_{at}$. The meter-out pressure $P_a$ with retraction working mode can be calculated by Equations (20) and (22), it is clear that the pressure $P_a$ is a constant when the meter-in pressure meter-in pressure $P_b$ is greater than the pressure $-P_{at}$, and the pressure $P_a$ is related to the variable parameter $F_L$ when the meter-in pressure meter-in pressure $P_b$ is less than the pressure $-P_{at}$.

## 4. Cavitation Performance and Prevention Potential Analysis

*4.1. Parameters Setting*

In this section, the calculations are carried out using MATLAB software to preliminarily evaluate the cavitation performances of HPCIMS and CLSS. To secure the effective meter-in and meter-out valve opening ratios for preventing cavitation, it is necessary to compare the cavitation performance of HPCIMS and CLSS with extension working mode and retraction working mode. The parameters are setting as follows:

(1). As listed in Table 1, all parameters of the two systems are standardized.

**Table 1.** The constant parameters of the two systems.

| Var. | Value | Units |
|:---:|:---:|:---:|
| $A_a$ | 0.0031 | m$^2$ |
| $A_b$ | 0.0015 | m$^2$ |
| $C_d$ | 0.7 | - |
| $W$ | 0.0023 | m |
| $x_{max}$ | 0.005 | m |
| $\rho$ | 850 | kg/m$^3$ |
| $\Delta P$ | 1 | MPa |
| $P_{at}$ | 0.101 | MPa |
| $P_c$ | 0.5 | MPa |

The head chamber area $A_a$ and rod chamber area $A_b$ are calculated by the cylinder-piston diameter and cylinder-rod diameter. The ratio $R$ is calculated by the $A_a$ and $A_b$. Assume that the vapor pressure $P_c$ is 0.5 MPa, which is greater than the atmospheric pressure $P_{at}$.

(2). The variable parameters of the two systems are listed in Table 2.

To obtain the general results, the external active load $F_L$ is increasing from 0 N to 40,000 N in a sine curve, as is shown in Figure 12. The ratio $\mu$ of HPCIMS is the $\mu_{min}$, which is calculated by the Equation (19) or (23). The ratio $\mu$ of CLSS is selected by four typical values 0.5, 1, 2 and 5. The meter-in valve opening ratio $x_{in}$ is increased proportionally from 0 to 1, and the meter-out valve opening ratio $x_{out}$ is calculated by the opening ratio $x_{in}$ and the ratio $\mu$.

**Table 2.** The variable parameters of the two systems.

| Var. | HPCIMS | CLSS | Units |
|:---:|:---:|:---:|:---:|
| $F_L$ | 0 to 40,000, Sine curve | 0 to 40,000, Sine curve | N |
| $\mu$ | $\mu_{min}$ | 0.5, 1, 2 and 5 | - |
| $x_{in}$ | 0 to 1 | 0 to 1 | - |
| $x_{out}$ | $x_{in}/\mu$ | $x_{in}/\mu$ | - |

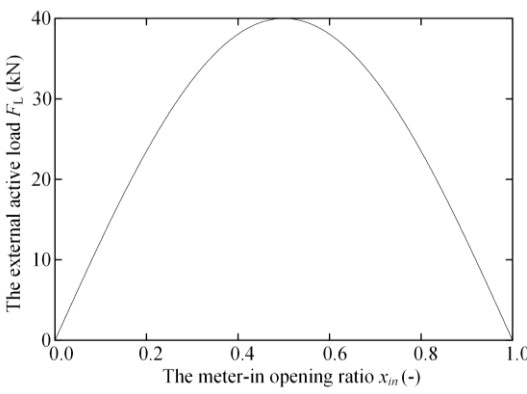

**Figure 12.** The external active load $F_L$ of the two systems.

### 4.2. Analysis Results of CLSS

As is shown in Figure 7, the proportional directional spool 5/3-valve of the CLSS is the main control valve, which is symmetric or asymmetric. If the main control valve is symmetric, the ratio $\mu$ equals 1. In most cases, the main control valve is asymmetric, so assume that the ratio $\mu$ equals four typical values 0.5, 1, 2, and 5. To obtain the general results, the extension working mode and retraction working mode are analyzed in this section.

Taking the Equations (13), (16)–(18) into calculations of CLSS with extension working mode, the analysis results can be seen in Figures 13–17.

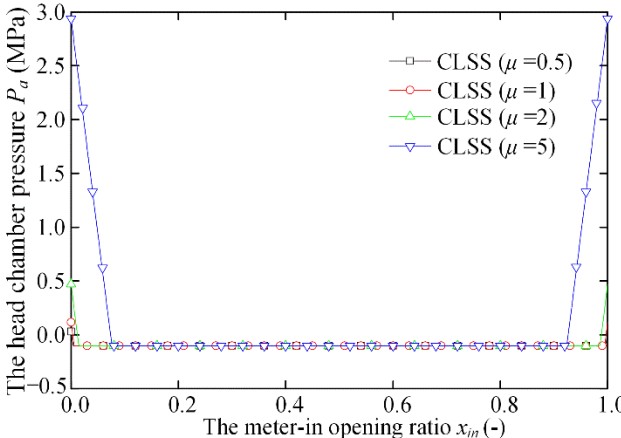

**Figure 13.** The head chamber pressure $P_a$ of CLSS with extension working mode.

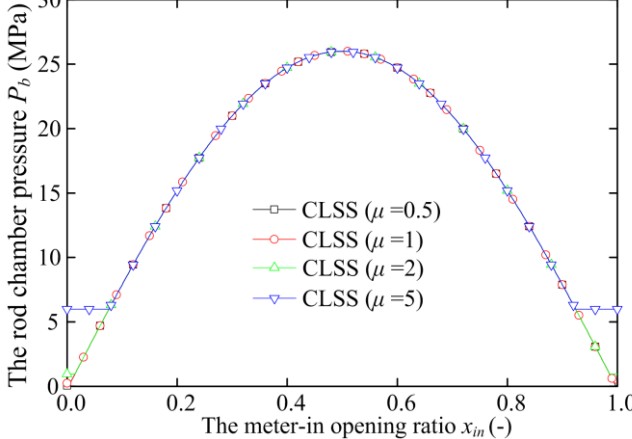

**Figure 14.** The rod chamber pressure $P_b$ of CLSS with extension working mode.

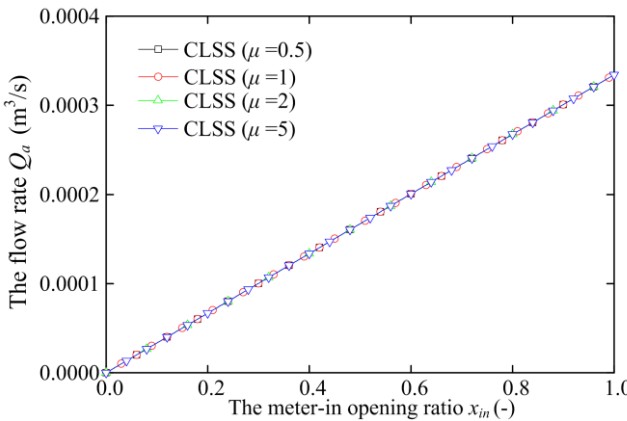

**Figure 15.** The flow rate $Q_a$ of the cylinder head chamber of CLSS with extension working mode.

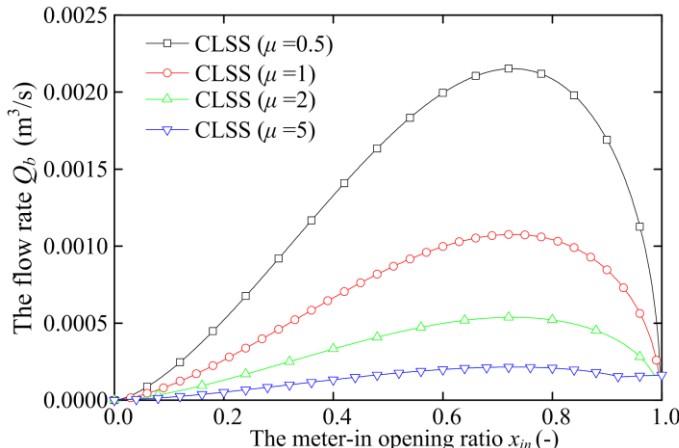

**Figure 16.** The flow rate $Q_b$ of the cylinder rod chamber of CLSS with extension working mode.

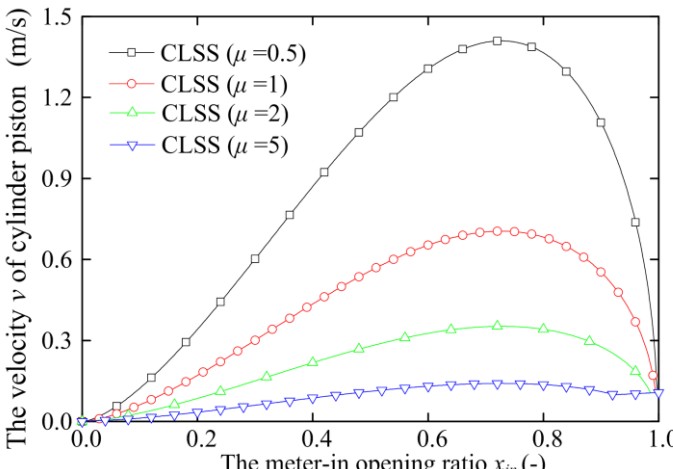

**Figure 17.** The velocity $v$ of the cylinder piston of CLSS with extension working mode.

Figure 13 shows that when the external active load $F_L$ is increasing at a certain value, the head chamber pressure $P_a$ is below gauge pressure 0 MPa when the meter-in opening ratio $x_{in}$ is generally from 0.1 to 0.9 no matter what the ratio $\mu$ is 0.5, 1, 2 and 5, and the minimum value of the ratio $\mu$ is −0.101 MPa which is the negative atmosphere pressure. Hence, it will be produced the cavitation phenomenon in the head chamber.

Figure 14 shows that the rod chamber pressure $P_b$ is changing, which is depending on the external active load $F_L$. Figure 15 shows that the flow rate $Q_a$ of the cylinder head chamber is proportional to the meter-in opening ratio $x_{in}$ because of the existence of the pressure compensator.

From Figures 16 and 17, it is clear that the flow rate $Q_b$ of the cylinder rod chamber and the velocity $v$ of the cylinder piston are changing nonlinearly with increasing of the meter-in opening ratio $x_{in}$, and the values of the $Q_b$ and $v$ are increasing-with-increasing of the ratio $\mu$. Compared with Figures 15 and 16, the flow rate $Q_a$ and $Q_b$ are not satisfied with the continuity equations. It is not desired that the velocity $v$ of the cylinder piston of the CLSS with extension working mode is nonlinear.

Similarly, taking the Equations (13), (16)–(18) into calculations of CLSS with retraction working mode, the analysis results can be seen in Figures 18–22.

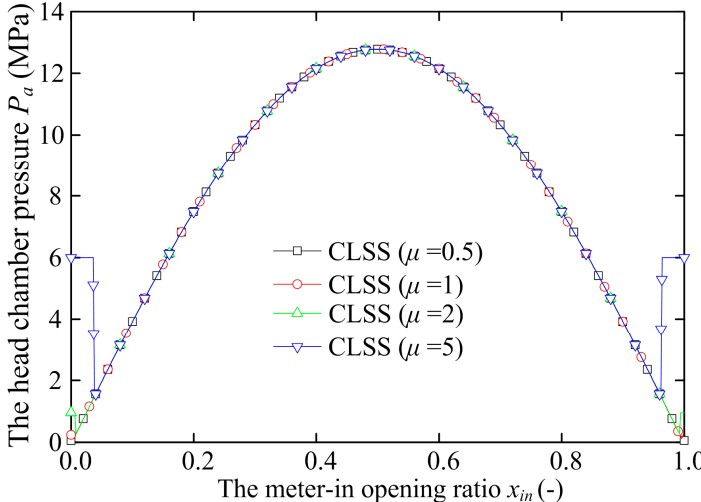

**Figure 18.** The head chamber pressure $P_a$ of CLSS with retraction working mode.

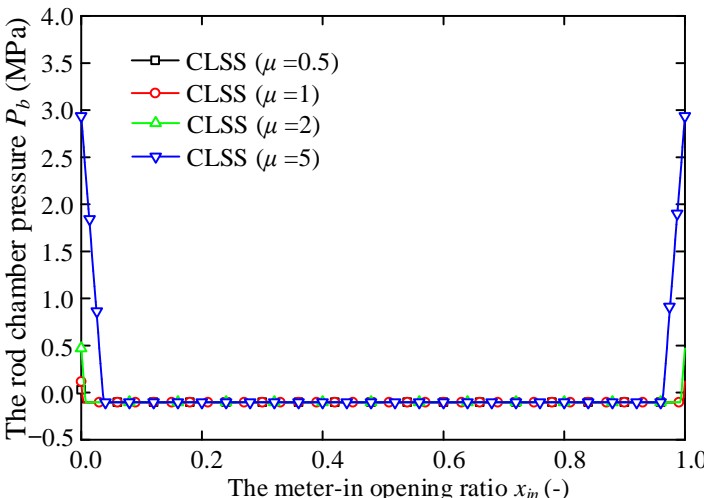

**Figure 19.** The rod chamber pressure $P_b$ of CLSS with retraction working mode.

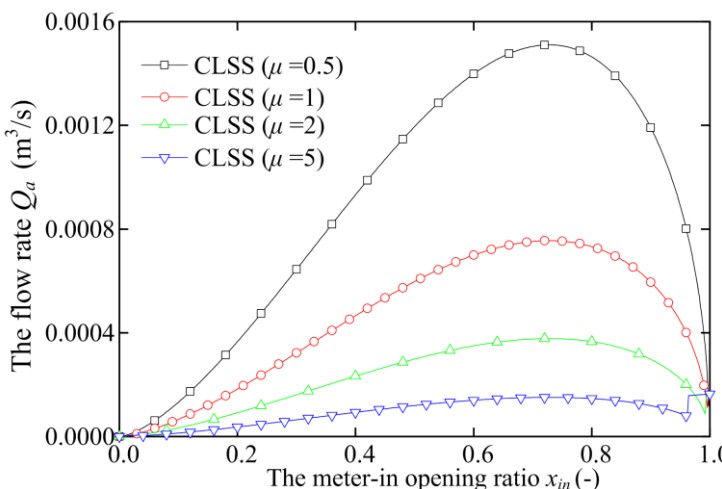

**Figure 20.** The flow rate $Q_a$ of the cylinder head chamber of CLSS with retraction working mode.

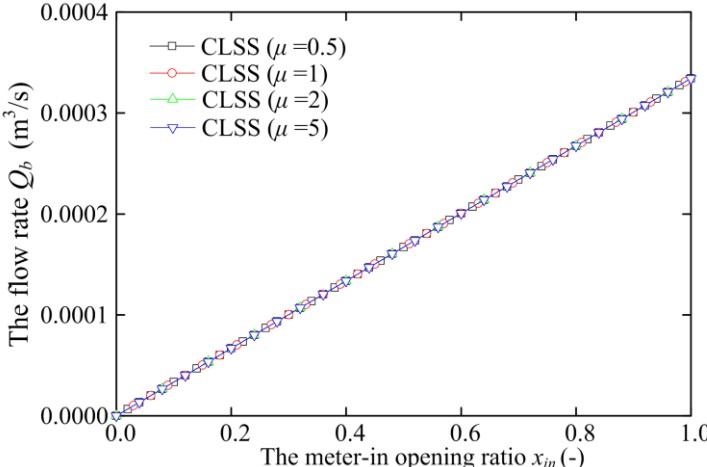

**Figure 21.** The flow rate $Q_b$ of the cylinder rod chamber of CLSS with retraction working mode.

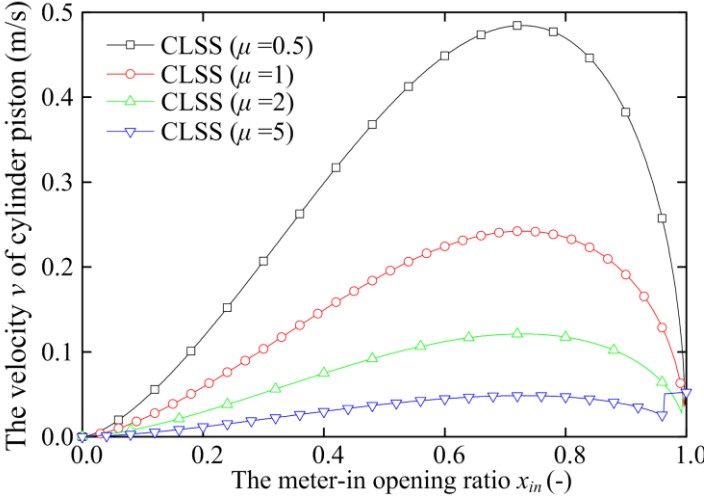

**Figure 22.** The velocity $v$ of the cylinder piston of CLSS with retraction working mode.

It is different from the extension working mode, the cylinder head chamber is the meter-out chamber, and the cylinder rod chamber is the meter-in chamber in the retraction working mode. Figure 18 shows that in the retraction working mode, the head chamber

pressure $P_a$ is changing as the variations of external active load $F_L$. Figure 19 shows that the rod chamber pressure $P_b$ is below gauge pressure 0 MPa when the meter-in opening ratio $x_{in}$ is generally from 0.05 to 0.95 no matter what the ratio $\mu$ is 0.5, 1, 2 and 5, and the minimum value of the ratio $\mu$ is $-0.101$ MPa. Hence, it will produce the cavitation phenomenon in the rod chamber. Figure 21 shows that the flow rate $Q_b$ of the cylinder rod chamber is proportional to the meter-in opening ratio $x_{in}$ because of the existence of the pressure compensator. Figures 20 and 22 show that the flow rate $Q_a$ of the cylinder head chamber and the velocity $v$ of the cylinder piston are changing nonlinearly with an increase of the meter-in opening ratio $x_{in}$. Compared with Figures 15 and 16, the flow rate $Q_a$ and $Q_b$ are also not satisfied with the continuity equations. It is not desired that the velocity $v$ of the cylinder piston of the CLSS with retraction working mode is nonlinear.

As a result, in the extension working mode or retraction working mode of CLSS, the meter-in chamber pressure of the cylinder will be below gauge pressure 0 MPa when the external active load $F_L$ is increasing at a certain value, so will be produced the cavitation phenomenon. The velocity $v$ of the cylinder piston in the extension working mode or retraction working mode of CLSS is nonlinear to the meter-in opening ratio $x_{in}$ or the meter-out opening ratio $x_{out}$ when the meter-in chamber pressure is below gauge pressure 0 MPa, and it is not desired for the hydraulic system. Because the ratio $\mu$ is selected by four typical values 0.5, 1, 2, and 5, the meter-out opening ratio $x_{out}$ is proportional to the meter-in opening ratio $x_{in}$.

### 4.3. Analysis Results of HPCIMS

In the HPCIMS, there are two control valves that can independently control the inflow rate and outflow rate of the cylinder chambers, so the ratio $\mu$, which is the ratio of the meter-in opening ratio $x_{in}$ and the meter-out opening ratio $x_{out}$ is changed, and it can be calculated by the Equation (19) or (23) for preventing cavitation phenomenon. Taking the Equations (12)–(23) into calculations of HPCIMS (hydromechanical pressure compensation independent metering system) with extension working mode and retraction working mode, the analysis results can be seen in Figures 23–28.

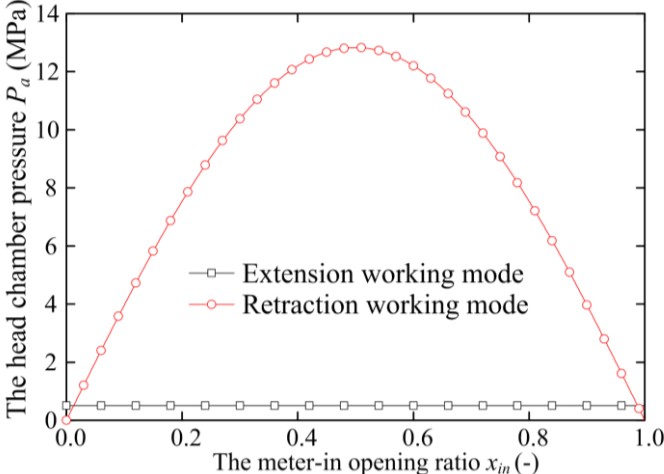

**Figure 23.** The variations of the head chamber pressure $P_a$ of HPCIMS.

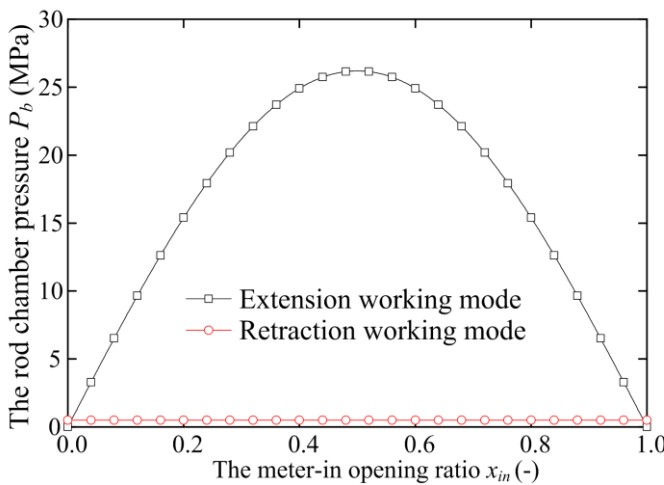

**Figure 24.** The variations of the rod chamber pressure $P_b$ of HPCIMS.

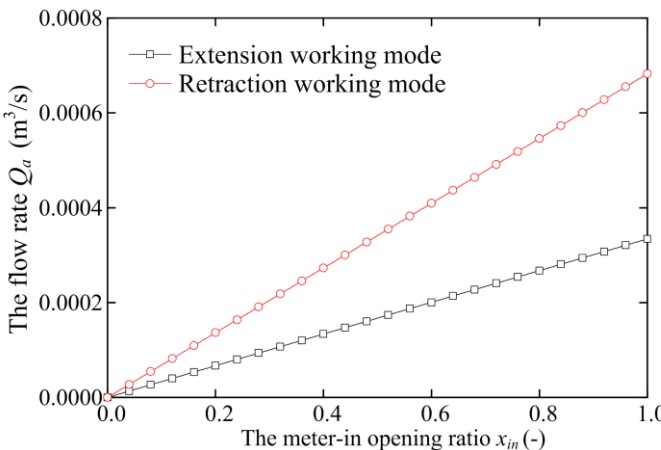

**Figure 25.** The variations of the flow rate $Q_a$ of the cylinder head chamber of HPCIMS.

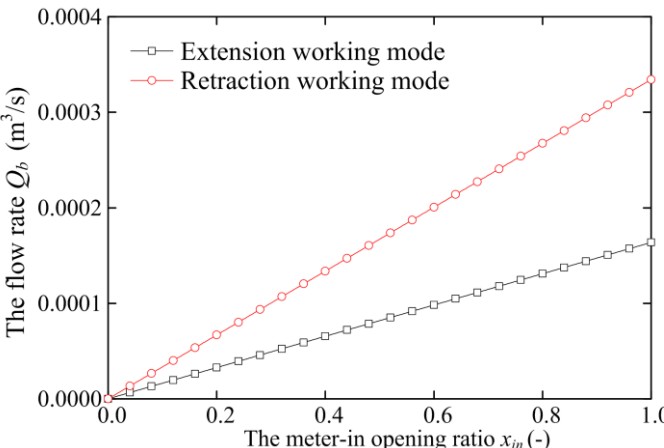

**Figure 26.** The variations of the flow rate $Q_b$ of the cylinder rod chamber of HPCIMS.

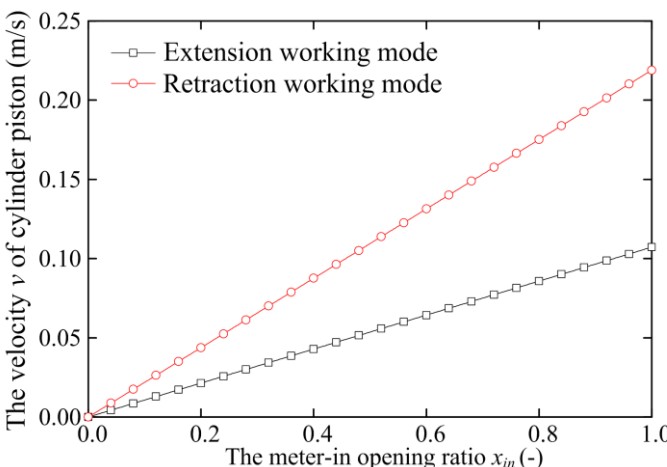

**Figure 27.** The variations of the velocity $v$ of the cylinder piston of HPCIMS.

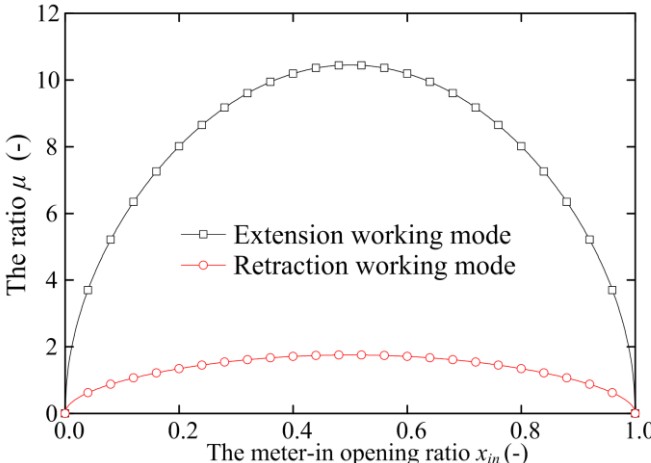

**Figure 28.** The variations of the ratio $\mu$ of HPCIMS.

Figures 23 and 24 show that the head chamber pressure $P_a$ of HPCIMS with extension working mode and the rod chamber pressure $P_b$ of HPCIMS with retraction working mode are stable nearby the vapor pressure $P_c$, which is setting as 0.5 MPa, and the head chamber pressure $P_a$ of HPCIMS with retraction working mode and the rod chamber pressure $P_b$ of HPCIMS with extension working mode is increasing by the external active load $F_L$.

As is shown in Figures 25 and 26, the flow rate $Q_a$ and $Q_b$ of HPCIMS with two working modes are proportional to the meter-in opening ratio $x_{in}$. Compared with Figures 25 and 26, the flow rate $Q_a$ and $Q_b$ are satisfied with the continuity equations.

In Figure 27, it is clear that the velocity $v$ of the cylinder piston of HPCIMS with two working modes is proportional to the meter-in opening ratio $x_{in}$. It is different from the CLSS, as is shown in Figure 28, the ratio $\mu$ of HPCIMS is changing with the meter-in opening ratio $x_{in}$.

As a result, the cavitation phenomenon in the meter-in chamber of CLSS when the external active load is inevitable. However, in HPCIMS, it can be preventing the cavitation phenomenon by changing the ratio $\mu$. To compare with the CLSS, the HPCIMS has a greater advantage for preventing cavitation phenomenon with external active load.

## 5. Conclusions

The working principle of HPCIMS and its cavitation performances was investigated in this paper. In the hydraulic system, the cavitation phenomenon occurs when the pressure falls below the vapor pressure. Compared with the CLSS, the meter-in opening

ratio $x_{in}$ and meter-out opening ratio $x_{out}$ of the HPCIMS can be regulated independently. When changing the ratio $\mu$ of the meter-in opening ratio $x_{in}$ and meter-out opening ratio $x_{out}$ can prevent cavitation phenomenon according to the derived equations. The cavitation performances of the CLSS and HPCIMS are compared by the calculations of the above equations.

In the CLSS with two working modes, the meter-in pressure will fall below gauge pressure 0 MPa when the external active load $F_L$ is increasing at a certain value. In HPCIMS with two working modes, the meter-in pressure can be kept as a fixed value which is greater than or equal to vapor pressure $P_c$ by changing the ratio $\mu$.

When the external active load $F_L$ is increasing at a certain value, the meter-in flow rate of CLSS is linear, the meter-out flow rate of CLSS is nonlinear, both the meter-in flow rate and meter-out flow rate of CLSS are not satisfied with the continuity equations. In HPCIMS, both the meter-in flow rate and meter-out flow rate are linear and are satisfied with the continuity equations. The velocity $v$ of the cylinder piston of the CLSS with two working modes are nonlinear. However, the velocity $v$ of the cylinder piston of HPCIMS with two working modes is proportional to the meter-in opening ratio $x_{in}$.

Hence, the cavitation phenomenon can be prevented in HPCIMS with the external active load by changing the ratio $\mu$ according to the derived equations. Moreover, it can be obtained the linear velocity $v$ of the cylinder piston with the meter-in opening ratio $x_{in}$.

In this study, the dynamic effects and system stability are ignored, so the analysis results have a certain limitation. The results are considered a reference for designing the software and hardware of HPCIMS. Moreover, the cavitation phenomenon preventing the control strategy of the HPCIMS yet requires further research.

**Author Contributions:** Investigation, K.L. and S.K.; methodology, S.K. and H.Q.; validation, K.L. and H.Q.; data curation, H.Q. and C.Y.; writing—original draft preparation, K.L.; writing—review and editing, S.K. and H.Q.; writing—analysis and conclusion, K.L., S.K. and H.Q.; supervision, S.K. and C.Y.; funding acquisition, K.L., S.K. and C.Y. All authors have read and agreed to the published version of the manuscript.

**Funding:** This research was supported by the Youth program of the National Natural Science Foundation of China (Grant No. 51805228), Natural Science Fund for Colleges and Universities in Jiangsu Province (Grant No. 20KJB580005) and the Social Development Project of Jiangsu Key Research and Development (Grant No. BE2018641).

**Institutional Review Board Statement:** Not applicable.

**Informed Consent Statement:** Not applicable.

**Data Availability Statement:** The data presented in this study are available on request from the corresponding author.

**Conflicts of Interest:** The authors declared no conflicts of interest.

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
