# Peer review of "Cavitation Prevention Potential of Hydromechanical Pressure Compensation Independent Metering System with External Active Load"

_processes, doi:10.3390/pr9020255_

Round 1

Reviewer 1 Report

Referee comments on the paper by Kailei Liu, Shaopeng Kang, Hongbin Qiang, and Chengtao Yu „ Cavitation Prevention Potential of Hydro-mechanical Pressure 2 Compensation Independent Metering System with External Ac-3 tive Load” presented for publication in the Processes

The article is written in a simple and transparent way. The subject of undertaken research is justified. There are no principal errors in the paper.

The theoretical introduction is written quite carefully, the literature analysis is quite thorough

But, I just have only a few suggestions:

Please check in the instruction if before quoting should be a space or not, in the article is different, e.g.

Line 31

is actuators[4]

and in line 27 or 29 there is no, e.g. applications [1,2].

Equation (1)

Line 173

Author write „flow coefficient Cd can be seen as constant values”, why and what was its value, was it the same in every case?

With these modification, the work will become suitable for publication Processes

Author Response

Responds:

We are very sorry about the format of the quoting in the instruction, we have removed the space before quoting as is shown in Red marked portion.

The description of the parameter W and Cd are revised as is shown in Red marked portion, and reference[26] is added.

The parameter W is the area gradient of the spool valve, which is determined by dimensions of the spool valve, so it can be seemed as a constant value when the spool valve is same.

The flow coefficient Cd equals 0.611 in the turbulent region[26], so it can be seen as a constant value because the turbulent flow is generally existed in the valve control system.

Reviewer 2 Report

Detailed comments are presented in the attached file

Author Response

Due to the limitations of the issue adopted by the authors, the volume of the presented work (23 pages) seems too large. In particular, the content of chapters 1, 2 and 3.1 should be given a more compact form. General comments: for figs. 1 - 11 please indicate sources of origin.

Responds: We have rearranged the chapters 1,2 and 3.1, and now the revision article has 21 pages. Figs. 1 – 11 are drew by ourselves according to the principles.

The derivation of the basic equations describing static and quasi-static states of both classes of systems, carried out in chapters 2 and 3, does not raise objections. However, it should be remembered that all the time the authors analyze the model burdened with numerous simplifications.

Responds: Yes, it is. So we have explained that this study has a certain limitation in Section 5 as is shown in Red marked portion.

Chapter 4 is the most important part of the work and how such a structure should be changed.The title of Chapter 4 should more adequately reflect the relationship between analysis and reduction of cavitation.

Responds:Yes, it is. We have rearranged the chapter 4 as is shown in Red marked portion.The title of Chapter 4 is Cavitation Performance and Prevention Potential Analysis.

The "serial" system adopted by the authors does not make it easier for the reader to compare the parameters of both of systems analyzed. In particular, the significant distance between the graphs (Fig. 12 - 28) and their interpretation in the text is burdensome.According to the reviewer, a "parallel" arrangement will be much more advantageous and clear for the reader: individual classes of characteristics for both types of arrangements are compared in one subchapter (similarly to Table 2). The interpretation and comparison of the created pairs of graphs should be relatively close to the text, which will greatly facilitate the reader's analysis.It is proposed to indicate clear benchmarking criteria for both classes of systems.

Responds: We are sorry about it, and we have rearranged the Chapter 4 for easy to read as is shown in Red marked portion.

Reviewer 3 Report

Dear authors,

Although the presented article seems very interesting to me, I regret to tell you that the current version of the article is not ready for publication and the overall text needs a major revision.

Attached to this file, you may find the PDF version of your article along with my comments embedded in the text. In addition to them, I have the following concerns:

1- In general, there is much basic information presented in the text. You may remove the extra and basic ones.

2- Your method, platform, assumptions, and the system that you might have been inspired by are not presented in the text.

3- Although the introduction seems reasonable in size, it does not really show where your work is located in the literature. I would like to have more literature review regarding the previous works by the others. 

4- The required references have to be added to the text.

5- Sometimes, the text does not seem consistent, and also the flow of the text is not maintained. Clearly speaking, the reader is surprised by the information which is presented and cannot find the reason why he/she is being fed by them and where in this study, they may come in handy.

6- English needs improvement. 

7- The article is in the form of a project report now. Please consider rebuilding it according to scientific writing principles.

8- And most importantly that lead to this decision of mine is the poor presentation of the results. Please go deep into your results and look at them from different perspectives and try to challenge yourself with questions. Present the required theory and principles then present your result and get rid of the other information. Please present one result, discuss and then move on to the other. It is easier to follow your arguments this way.

All in all, I wish you luck with the comments and suggestions that I left for you. I hope they are useful to improve the quality of the text and prepare it for publication.

Author Response

1- In general, there is much basic information presented in the text. You may remove the extra and basic ones.

Responds:We have removed the extra and basic information presented in the text.

2- Your method, platform, assumptions, and the system that you might have been inspired by are not presented in the text.

Responds:The main content of this paper is cavitation performance analysis and cavitation prevention method of the HPCIMS (hydro-mechanical pressure compensation independent metering system). The method of this paper is formula derivation and theoretical analysis of the HPCIMS. The platform is the computer calculation platform. The assumptions are presented in Section 3 and Section 4, the fluid compressibility and the dynamic effects are also ignored, and all parameters of the two systems are standardized. The system is HPCIMS which is presented in Section 2.

3- Although the introduction seems reasonable in size, it does not really show where your work is located in the literature. I would like to have more literature review regarding the previous works by the others. 

4- The required references have to be added to the text.

Responds:The main research literatures of the independent metering system have been listed. However, the research literature which is related to the cavitation performance analysis and prevention method of the independent metering system is very rare. In the Introduction of the revison, the sentence “Many scholars have studied the independent metering system for a long time, but the researches related to the cavitation performance analysis of the independent metering system is still very rare.” is added as is shown in Red marked portion.One reference is added to the text.

5- Sometimes, the text does not seem consistent, and also the flow of the text is not maintained. Clearly speaking, the reader is surprised by the information which is presented and cannot find the reason why he/she is being fed by them and where in this study, they may come in handy.

Responds:We are very sorry about it. We have rearranged and corrected the article as is shown in Red marked portion. 

6- English needs improvement. 

Responds:We are very sorry about it, and we corrected the article as is shown in Red marked portion.

7- The article is in the form of a project report now. Please consider rebuilding it according to scientific writing principles.

Responds:We are very sorry about it, and we have rearranged and corrected the article.

8- And most importantly that lead to this decision of mine is the poor presentation of the results. Please go deep into your results and look at them from different perspectives and try to challenge yourself with questions. Present the required theory and principles then present your result and get rid of the other information. Please present one result, discuss and then move on to the other. It is easier to follow your arguments this way.

Responds:We are very sorry about it, and we have rearranged the Section 4 Comparison Results.

Round 2

Reviewer 2 Report

The manuscript has undergone significant rewrites.
The changes made to the second version of the manuscript improved its clarity. Although they have not incorporated all of the reviewer's suggestions verbatim, the quality of the justification for the thesis has significantly improved.

Author Response

Thank you very much for your careful review and constructive suggestions with regard to our manuscript. Those comments are helpful for authors to revise and improve our paper.

Reviewer 3 Report

Dear authors,

Thank you for revising the manuscript and responding patiently to my comments.

I have a few suggestions and a question for you:

1) Please consider grammar check by available toolboxes such as Grammarly for your manuscript. I could also detect some typo that Grammarly can also help you with these.

2) In the caption of the figures, although stated in detail in the text, try to describe all the components in the figure. It should be in a way that if a reader just reads the caption of a figure, can realize what the details of the figure is.

3) And the question is, what do you mean by computer calculation platform? Please indicate if you used any application package or specific programming language. I suspect MATLAB from the figures in the results but cannot be sure.

Please specify clearly what programming language and/or package you used for the calculations and modeling.

After addressing the mentioned comments, my decision is acceptance. You do not need to send the manuscript back to me as what I mentioned here is just suggestions.

All in all, I wish you luck in your future endeavors and hope to read more from you.

Best regards,

Moein

Author Response

1) Please consider grammar check by available toolboxes such as Grammarly for your manuscript. I could also detect some typo that Grammarly can also help you with these.

Responds:

We are very sorry about the grammatical mistakes in English, we have used Grammarly for our manuscript as is shown in the revision manuscript.

 2) In the caption of the figures, although stated in detail in the text, try to describe all the components in the figure. It should be in a way that if a reader just reads the caption of a figure, can realize what the details of the figure is.

Responds:

We have revised the caption of some figures as is shown in the revision manuscript.

3) And the question is, what do you mean by computer calculation platform? Please indicate if you used any application package or specific programming language. I suspect MATLAB from the figures in the results but cannot be sure.

Please specify clearly what programming language and/or package you used for the calculations and modeling.

Responds:

We have used MATLAB software for calculating the results. In the Section 4, the sentence “In this section, the calculations are carried out using MATLAB software to preliminarily evaluate the cavitation performances of HPCIMS and CLSS.” is added as is shown in the revision manuscript.